# Effect of Microstructure on Mechanical Properties of 316 LN Austenitic Stainless Steel

**Kewei Fang \*, Kunjie Luo and Li Wang**

Suzhou Nuclear Power Research Institute, Suzhou 215004, China
\* Correspondence: fangkewei@cgnpc.com.cn

**Abstract:** The microstructure development of 316 LN austenitic stainless steel (316 LNSS) during the aging process is investigated in this article. The thermal aging processes were conducted at 750 °C with different periods ranging from 50 to 500 h. The metallographic results show that the coherent and incoherent twins were present in the original 316 LNSS grains, but dwindled as the aging period increased. After 50 h of aging, many fine, dispersed particles precipitated from the matrix, which were identified as $M_{23}C_6$ by energy dispersive spectrometer (EDS) and transmission electron microscopy (TEM). Additionally, the impact toughness and Brinell hardness (HBW) changed during the aging, which was closely related to the effects of dispersion strengthening and solution strengthening. A negatively linear relationship between Brinell hardness and Charpy impact energy was established, which could be utilized to predict the degree of thermal embrittlement.

**Keywords:** 316 LNSS; Charpy impact; HBW hardness; precipitates; twin structure





## 1. Introduction

There is an increasing desire in the electric utility business to reduce energy usage and $CO_2$ emissions in power plants. Most nations are constructing large-capacity ultra-supercritical (USC) units or advanced nuclear power plants to improve heat efficiency. The economic efficiency of generating units may be greatly improved by raising the steam temperature of USC to 600 °C. Meanwhile, to satisfy the needs of such harsh operating conditions, materials with enhanced heat resistance are necessary [1–3]. A severe accident pressure relief valve is also provided in European advanced pressurized water reactors to guarantee that the first-circuit pressure relief is executed easily in the case of a serious accident in a high-pressure molten reactor [4,5]. When the core outlet temperature hits 650 °C, it is manually opened. The design specifies for the valve body's impact toughness $KV_2$ to be more than 80 J/cm$^2$ at 750 °C after aging for 100 h at room temperature to assure the valve's long-term efficacy and reliability.

Because of its strong combination of enhanced creep strength [6], oxidation resistance [7], and hot corrosion resistance [8], 316 LN austenitic stainless steel (316 LNSS) has been acknowledged as one of the optimum materials for nuclear power, biomedical, and other industries [9–11]. He et al. [12] used tensile experiments to investigate the thermoplastic and high-temperature tensile fracture behavior of 316 LNSS at 850–1300 °C, and the results revealed that dynamic recrystallization occurred in this stainless steel during thermal deformation above 1000 °C, but the tensile fracture behavior of 316 LNSS at 750 °C was not studied. The thermal simulation testing equipment for 316 LNSS was explored by Jia et al. [13]. The findings revealed that dynamic strain aging occurs in 316 LNSS at high temperatures, but no relationship was found between the microstructure and dynamic strain aging behaviors. Samuel et al. [14] looked at the influence of temperature and strain rate on the tensile strength and section shrinkage of 316 LNSS at 750 °C, but they did not look at the effect of strain rate on the mechanical characteristics of the material. McQueen et al. [15] investigated the dynamic recrystallization behavior of 316 LNSS during

thermal deformation and found that dynamic reversion and dynamic recrystallization can soften the metal and improve its plasticity, but they did not investigate the deformation behavior or microstructure evolution of 316 LNSS. The majority of the abovementioned study topics are focused on the thermal deformation and dynamic recrystallization behaviors of 316 LNSS at high temperatures, and the findings have some guiding relevance for the forging process and 316 LNSS optimization. Meanwhile, there is currently a scarcity of studies into the development of precipitates in 316 LNSS at high temperatures. Furthermore, the current focus of 316 LNSS application performance research is on the organization's long-term thermal aging process at 300–400 °C base material and weld temperatures [16–18], with no mention of the effect of brief aging at 750 °C on its structure and attributes. Because carbides coarsen at grain boundaries in austenitic steel with a high Cr content, they are unfavorable for mechanical properties [19,20]. As a result, this paper uses 316 LNSS as the research object, and conducts thermal aging tests at 750 °C for various times, observes and analyzes the microstructure after aging, and provides an experimental basis for the development and promotion of austenitic stainless steel based on the impact toughness and Brinell hardness properties of 316 LNSS.

## 2. Materials and Experimental Procedures

### 2.1. Testing Material

The test material 316 LNSS (DIN 1.4429) was obtained from a nuclear power plant pipeline. The chemical composition of the 316 LNSS was studied using a full-spectrum direct-reading plasma spectrometer (PerkinElmer OPTIMA 2100DV, Waltham, MA, USA), a high-frequency infrared carbon and sulfur analyzer (Beijing Wanliandaxinke Instrument Co., Ltd. CS-902G, Beijing, China), and an oxygen, nitrogen, and hydrogen analyzer (Bruker G8 GALILEO, Karlsruhe, Germany). The chemical composition of the 316 LNSS was determined as follows (mass fraction, %): C 0.028, Si 0.050, Mn 1.78, P 0.021, S 0.001, Cr 17.28, Ni 12.48, Mo 2.55, Cu 0.087, Co 0.031, N 0.066, and the balance of Fe. The casting billets were forged into $230 \times 230$ mm forgings at 1200 °C forging start and 1000 °C forging end temperatures. Following forging, the billets were immediately treated to a solution treatment (solution treatment process: 1040 °C, 3 h holding time, air cooling out of the furnace).

### 2.2. Experimental Methods

The wire-cut $15 \times 15 \times 100$ mm specimens were vacuum sealed with quartz tubes and aged at 750 °C for 50 h, 100 h, 200 h, 300 h, and 500 h, respectively. To generate a metallographic sample, a $15 \times 15 \times 6$ mm piece was cut from the aging sample, mechanically polished, and then etched for 60 s with a ferric chloride hydrochloric acid solution (2.5 g $FeCl_3$ + 25 mL HCl + 50 mL $H_2O$). The samples were inspected using an optical microscope (OM, Zeiss Axiovert 200 MAT, Oberkochen, Germany), and the precipitated phases were evaluated using a scanning electron microscope (SEM, Zeiss Merlin Compact) and spectrometer (Oxford Inca, Oxford, UK), with the content of the precipitated phases being measured using the grid technique.

After aging, the sample was cut into thin slices of 0.2 mm thickness, mechanically ground to 0.15 mm, and cut into 3 mm discs, which were placed in the specimen holder of a double-spray apparatus equipped with platinum wire and metallographic electrolytic double-spray thinned until perforation, with an electrolytic etching solution of 80% methanol + 20% ($v/v$) nitric acid, cooled by liquid nitrogen, and operated at 15 V. Transmission electron microscopy (TEM, JEM-2100, Tokyo, Japan) was used to investigate the specimen films. A typical Charpy impact sample ($10 \times 10 \times 55$ mm) was cut from the aged specimen to determine the room temperature impact energy of 316 LNSS after various aging durations, and the test was repeated three times for each condition. Subsequently, the specimens' Brinell hardness (HBW) was determined.

## 3. Results and Discussion

### 3.1. Microscopic Structures Observation under OM

The OM images of the original and aged 316 LNSS specimens are shown in Figure 1. The metallographic organization is conventional austenite, with grain sizes ranging from 4 to 5 microns. There is practically no carbide precipitation at the grain boundaries and inside the grains due to the low carbon concentration. Meanwhile, in the grains of the original 316 LNSS material, twin structures may be seen, and they can be separated into coherent twins and incoherent twins. These twins originated during the annealing heat treatment [8,21] (i.e., solution treatment), which relies on the stacking fault slip [22] for nucleating and expanding. The grain sizes have not changed substantially after 50 h of aging, and the coherent and incoherent twins are still present in the 316 LNSS metallurgical structure (see Figure 1b). In addition, as seen in Figure 1b–f, the intragranular twins gradually fade away, and the twin boundaries become blurry. Incoherent twins have already become impossible to detect after 100 h of aging, while the coherent twin's interface has disappeared (see Figure 1c). The coherent twins fade with aging time, and after 500 h of aging, all twin borders are impossible to identify. The migration and degeneration of the stacking fault throughout the aging process might be the major cause of the twin structures disappearing [23]. The 316 LNSS microstructure shows no visible grain coarsening after 500 h of aging, indicating that the austenite grain structures are highly robust during thermal aging at 750 °C. Furthermore, a small number of second-phase particles [24] (black dots) are scattered throughout the grains of 316 LNSS (see Figure 1b–f). However, the properties of precipitates and their evolution behaviors cannot be observed using an optical microscope due to resolution constraints.

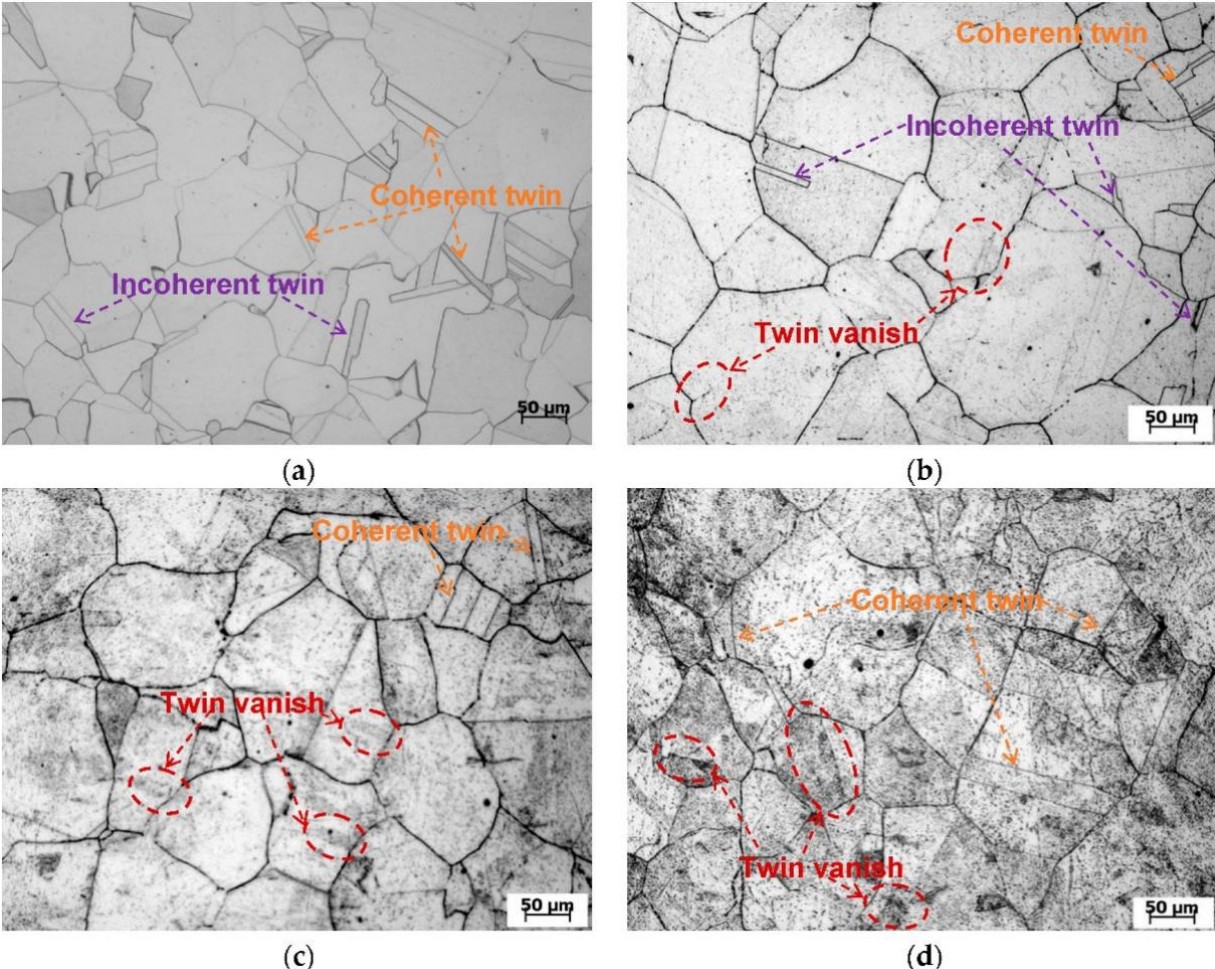

**Figure 1.** *Cont.*

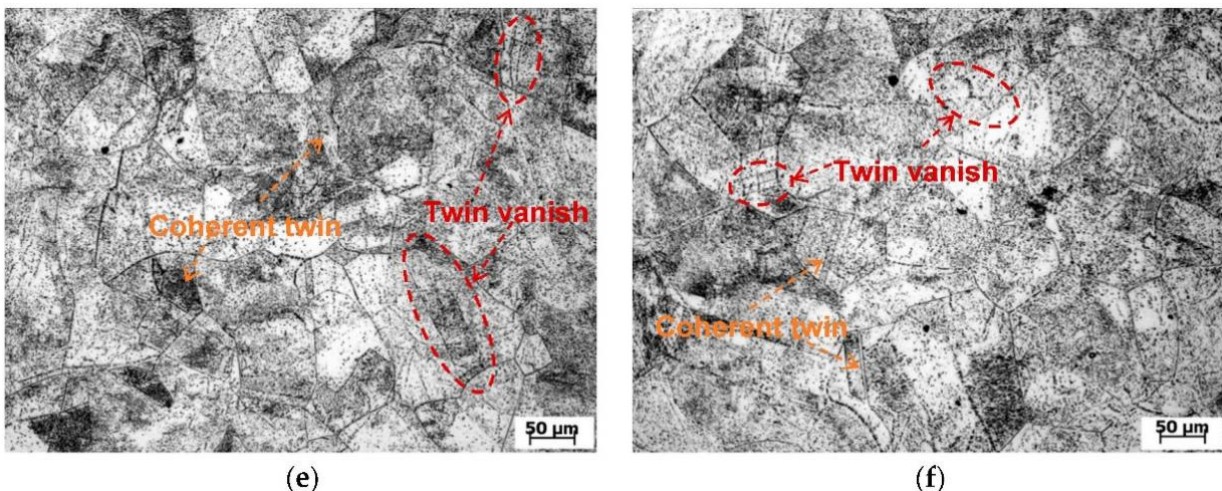

(**e**)                                                                 (**f**)

**Figure 1.** Metallurgical structure observations of 316 LNSS during thermal aging: (**a**) origin, (**b**) 50 h, (**c**) 100 h, (**d**) 200 h, (**e**) 300 h, and (**f**) 500 h.

*3.2. Microscopic Structures Observation under SEM*

SEM observations analyses are used to determine the evolution behaviors of precipitates in 316 LNSS during aging. Figure 2a shows a SEM picture of the original 316 LNSS. Before aging, there are hardly any precipitates in the grains or grain boundaries of the 316 LNSS microstructure.

However, after 50 h of aging, numerous small primary particles have precipitated from the grain boundaries, as illustrated in Figure 2b. A minor amount of short rod-like precipitated phases of various sizes have precipitated within the crystal after 100 h of aging. In addition, the number of precipitated phases at the grain boundaries also grows, and the distribution becomes more chain-like. Inter-granular precipitates, on the other hand, are larger and more prone to occur near grain boundary or triple grain boundary junctions [25]. In previous research, the grain boundary junctions are determined to be a kind of planar defect with a high lattice-distortion energy [26]. As a result, the carbon atoms at this position dispersed more easily, allowing for a combination reaction with other elements. The tendency of the film-like dispersion of precipitated phases at the grain boundary becomes more noticeable, even at some co-granular twin boundaries. This is an example of the normal Ostwald Ripening [27], which is a spontaneous energy transfer. Kinetically, small particles are preferred, whereas thermodynamically, large precipitates are preferred. Because smaller precipitates nucleate more easily, many tiny particles precipitate in the 316 LNSS at first (see Figure 2c,d). Such small particles, on the other hand, have a higher surface area-to-volume ratio, making them less energetically stable [28]. As a result, certain microscopic particles become larger (i.e., with a higher volume-to-surface area ratio), reducing their energy level. Figure 2 displays the complete process of precipitate behavior evolution in 316 LNSS as it ages.

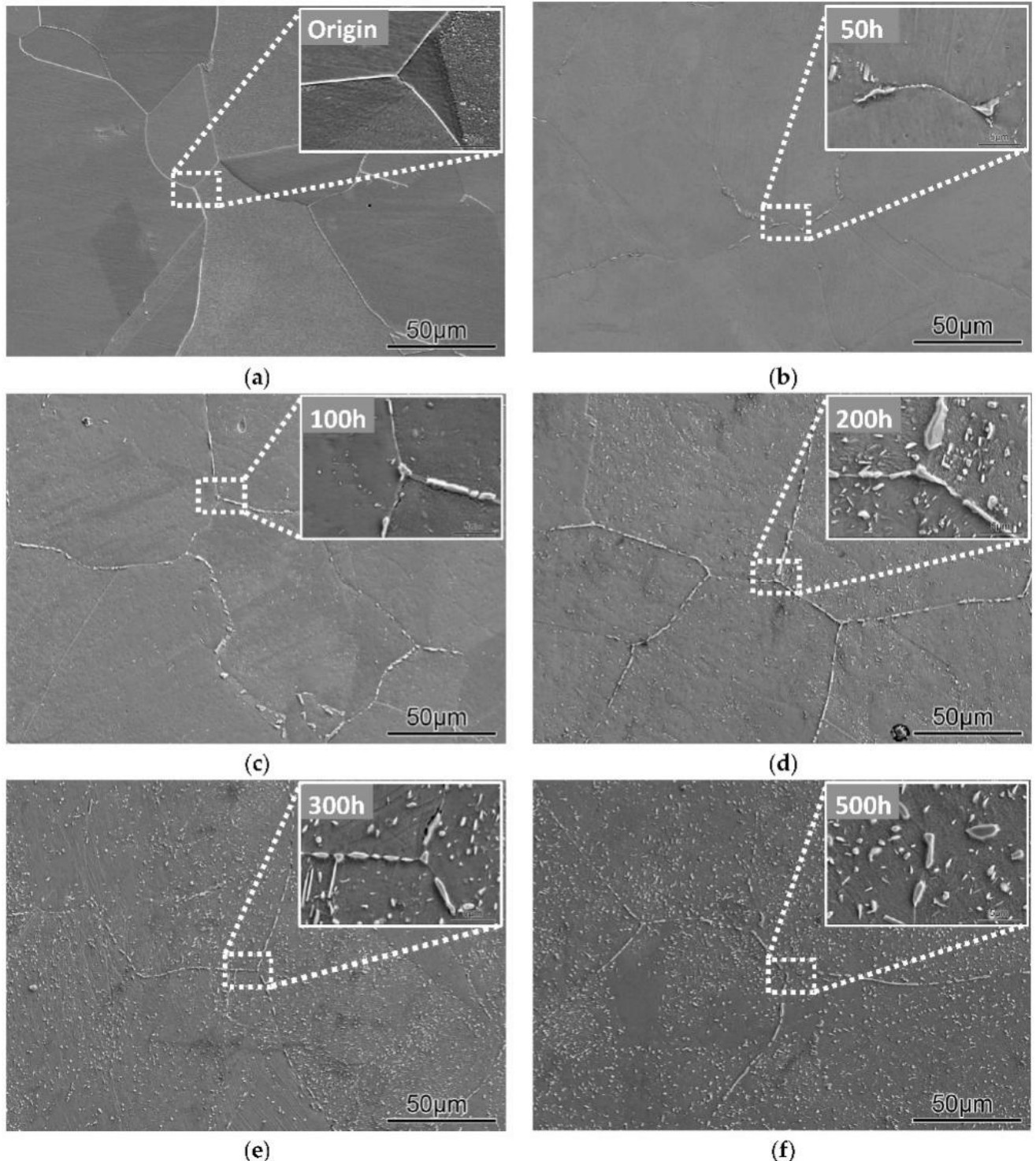

**Figure 2.** SEM observations of 316 LNSS during thermal aging: (**a**) origin, (**b**) 50 h, (**c**) 100 h, (**d**) 200 h, (**e**) 300 h, and (**f**) 500 h.

The intracrystalline and grain boundary-precipitated phases are both Cr- and Mo-rich $M_{23}C_6$ phases, according to SEM-EDS and TEM examination of the precipitates from the 300 h aged specimens [23] (see Figures 3 and 4).

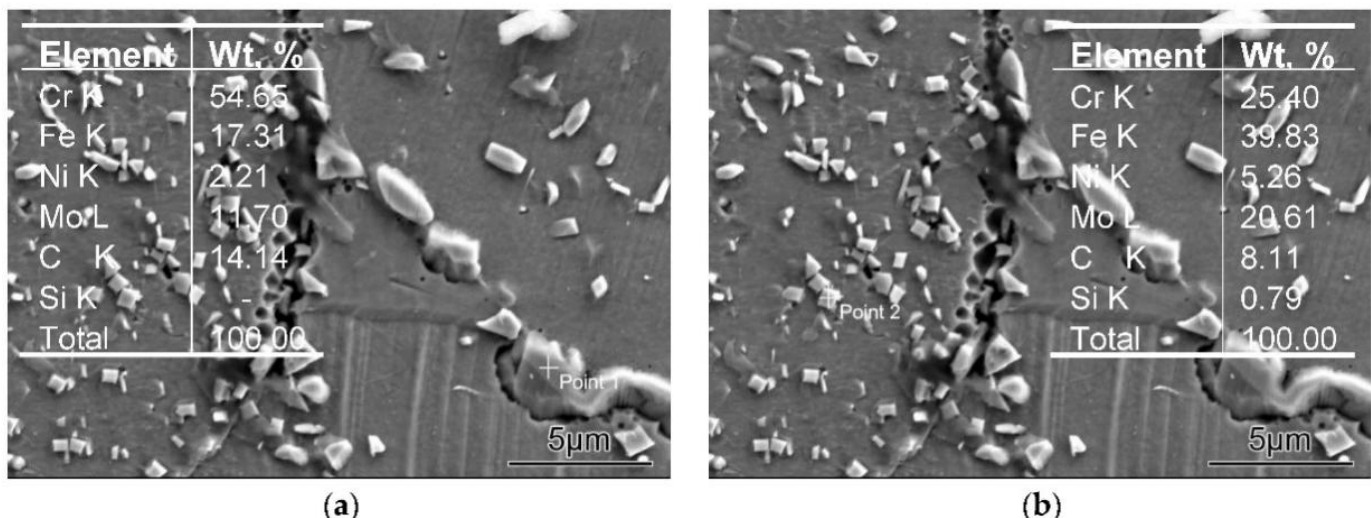

**Figure 3.** EDS analysis of the precipitates from the 300 h aged specimens: (**a**) intergranular precipitates, (**b**) intragranular precipitates.

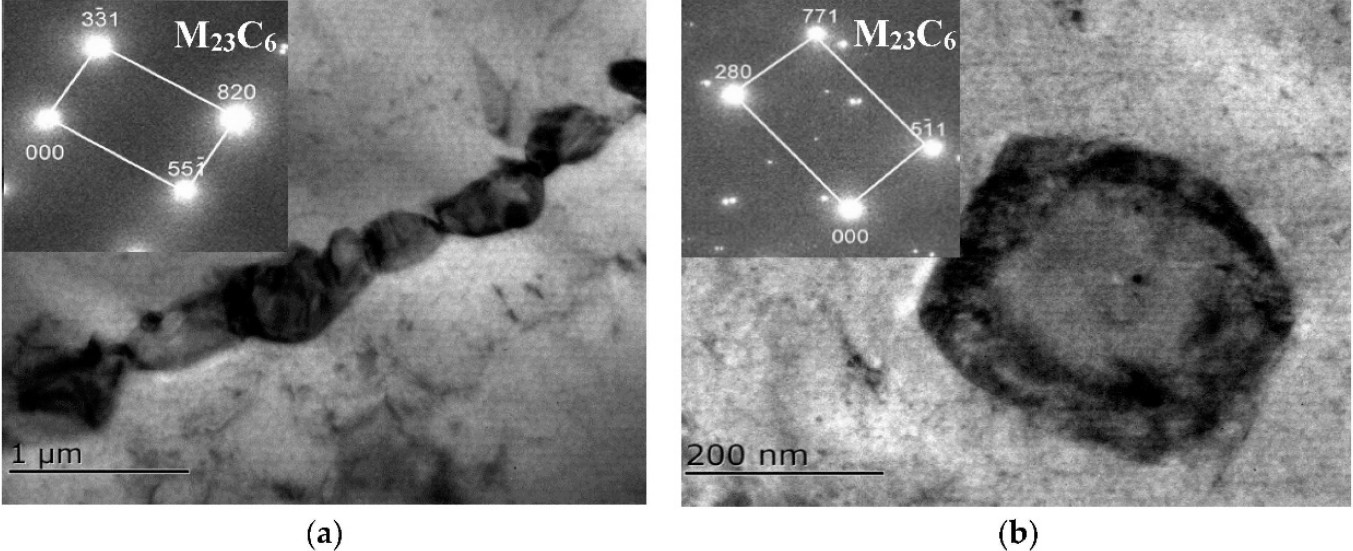

**Figure 4.** TEM analysis of the precipitates from the 300 h aged specimens: (**a**) intergranular precipitates, (**b**) intragranular precipitates.

The statistics of the area fraction of precipitates were done on the SEM images to analyze the evolution of precipitates in 316 LNSS in a quantitative manner. In each aging specimen, more than five distinct viewing areas were chosen at random. The total area of the precipitates was divided by the area of the field of view to obtain the area fraction of the precipitates [7,29–31]. The fluctuation of the area fraction of the precipitates in 316 LNSS with aging is shown in Table 1. The area proportion of the precipitates grew as the aging period increased, but the trend eventually stabilized once the time of effectiveness reached 300 h. The precipitation change rule was consistent with the prior SEM observations [32].

**Table 1.** Volume fraction of the precipitates.

| Aging Treatment | Volume Fraction (/%) |
|---|---|
| Origin | 0 |
| 750 °C 50 h | 3.19 |
| 750 °C 100 h | 7.89 |
| 750 °C 200 h | 13.03 |
| 750 °C 300 h | 18.16 |
| 750 °C 500 h | 19.09 |

*3.3. Charpy Impact and HBW Hardness Test*

Figure 5 illustrates the Charpy impact toughness and HBW hardness of 316 LNSS at room temperature after aging at 750 °C for various durations. The room temperature Charpy impact energy of 316 LNSS decreases continuously with the prolongation of aging time, and the decrease is larger in the early aging period. Then, the decrease of Charpy impact energy slows down, and the Charpy impact energy is essentially stable after 300 h of aging, as shown in Figure 5. After 500 h of thermal aging at 750 °C, the impact energy value has fallen to 13.7%. The Charpy impact toughness of the 316 LNSS decreases due to the material's hardening after aging, which causes it to shatter more easily on impact. The Charpy impact fracture morphology of 316 LNSS after aging at various time periods is shown in Figure 6. As illustrated in Figure 6a, the original specimen's impact fracture contains a high number of ripped tough nests, and the size of the tough nests is not uniform, indicating clear ductile fracture features. When aged at 750 °C for 50 h, the number of tough nests decreased and a tiny amount of grain boundaries developed on the fracture, compared to the initial condition [33]. The plastic fracture feature with low grain boundary fraction could be attributed to the intergranular carbides. Under this condition, the carbide concentration was only approximately 3.19%, as presented in Table 1. The number of tough nests decreased when the aging time was raised to 100 h, and a high number of grain boundaries emerged, demonstrating a mixture of ductile and brittle fractures, as shown in Figure 6c. This was attributed to the increased aging time and the rise in carbide content to 7.89% [34]. The carbide at the grain boundaries grew to 13.03% when the aging time was increased to 200 h, and the intergranular rupture revealed a brittle fracture in the shape of icing sugar, with tiny and shallow tough nests dispersed over the crystal surface [35], as shown in Figure 6d. The tough nests nearly vanished completely as the aging time rose to 300 and 500 h, and the icing sugar-like structures became more visible (Figure 6e–f), since the intergranular carbide concentration was approximately 19% at this time.

The HBW hardness and Charpy impact energy change in the opposite direction: as the aging time is extended, the HBW value continues to rise, and the rise is greater in the early aging period. As the aging time is extended, the rise of the HBW value tends to slow [36]. After 300 h of aging, the HBW value is essentially stable. The HBW value rose by 21.9% after 500 h of aging.

There is an inherent relationship between Charpy impact energy and HBW, according to the Charpy impact energy trend. Figure 7 depicts the linear and negatively correlated connection between Charpy impact energy and hardness (HBW), with their values fitted using the relationship equation:

$$A_k = 1548.12 - 8.35 \times \text{Hardness} \tag{1}$$

where $A_k$ is the Charpy impact energy, $J/cm^2$, and Hardness represents the Brinell hardness value, HBW.

The 316 LNSS to Brinell hardness equation is used to compute the room temperature Charpy impact energy, which is used to forecast the degree of thermal embrittlement. The service temperature and time for 316 LNSS that is already in use are known, and the Brinell hardness value measured by the material may be used to forecast the Charpy impact energy values at that time to determine the degree of thermal embrittlement of

the material. It should be noted that the grain size of 316 LNSS also expands at 750 °C in long-term service, which also influences its impact toughness. This effect cannot be directly expressed by Brinell hardness. The authors will, therefore, conduct more in-depth study on the influencing elements in this respect.

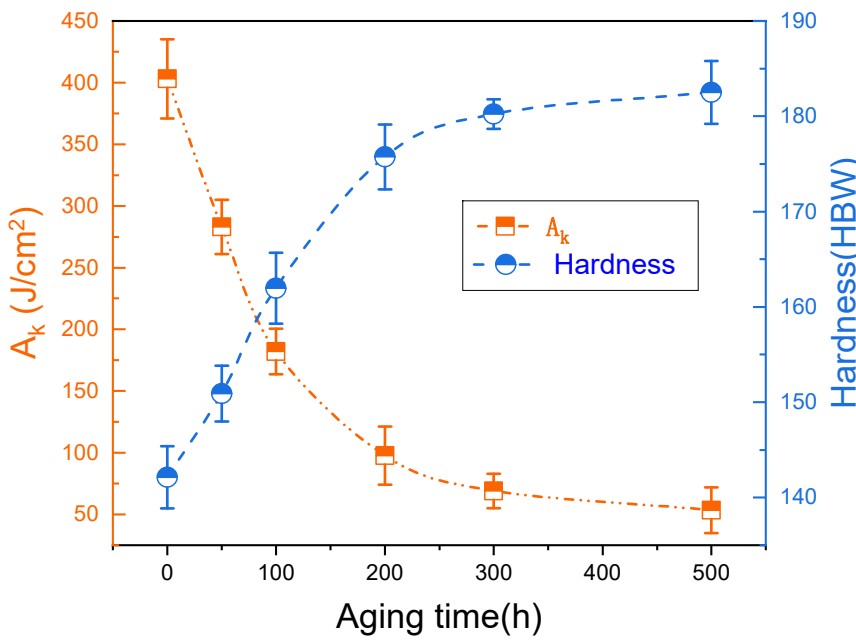

**Figure 5.** Charpy impact and hardness test result of 316 LNSS at 750 °C aging.

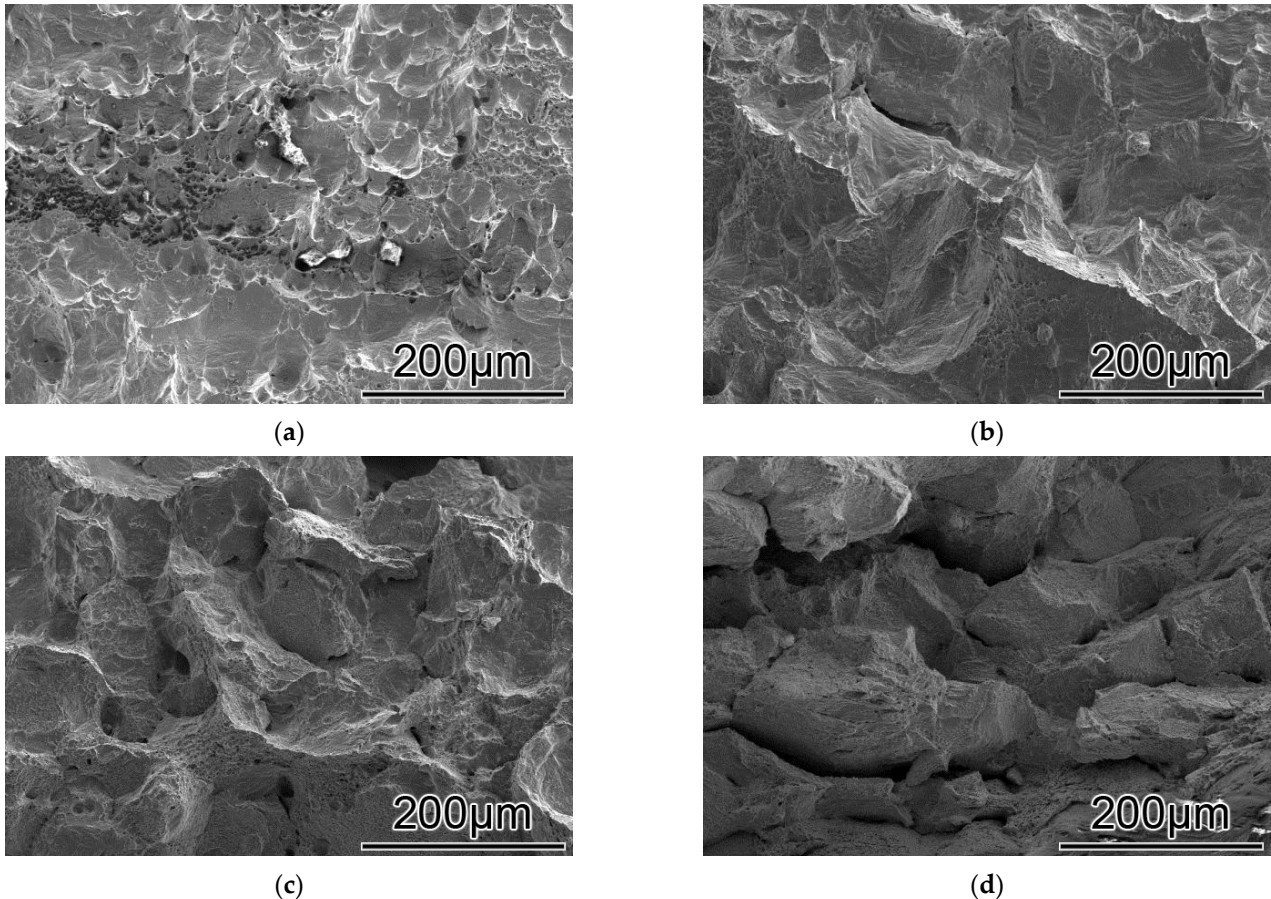

**Figure 6.** *Cont.*

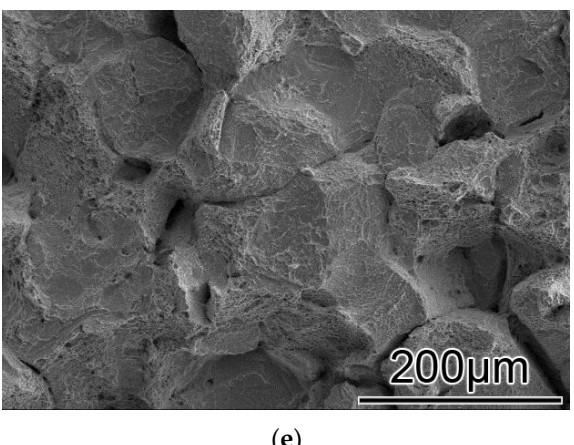
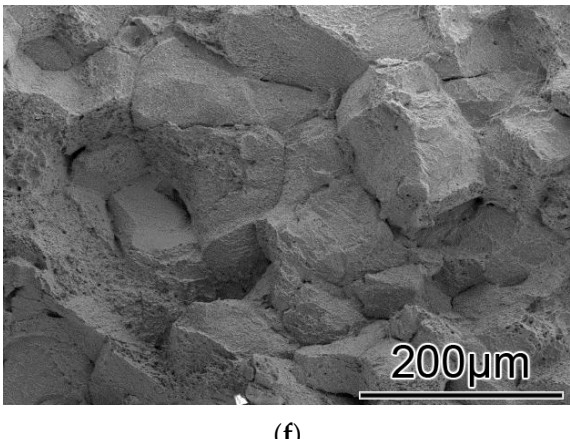

(**e**)　　　　　　　　　　　　　　　　　　　(**f**)

**Figure 6.** Fracture analysis of the specimen after impact test: (**a**) origin, (**b**) 50 h, (**c**) 100 h, (**d**) 200 h, (**e**) 300 h, and (**f**) 500 h.

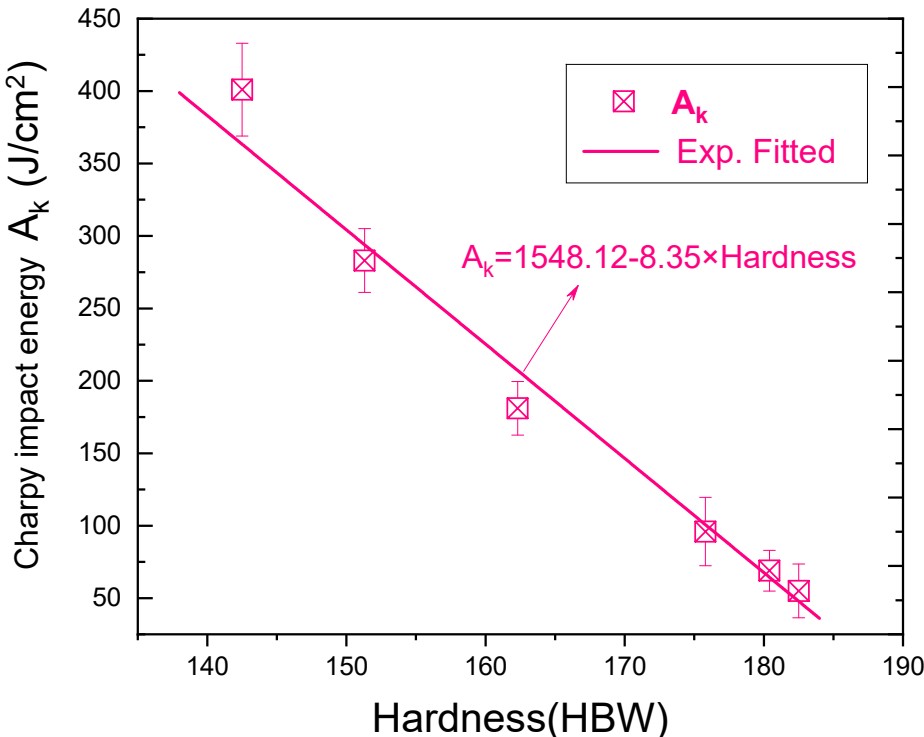

**Figure 7.** Relationship between Charpy impact energy and Brinell hardness value after aging at 750 °C.

## 4. Conclusions

The development of precipitates in 316 LNSS after aging at 750 °C was examined in this work. The Charpy impact characteristics and Brinell hardness were measured before and after aging. The following is a summary of the key conclusions:

1.  In grains of the original 316 LNSS specimens, coherent twins and incoherent twins exist. The intragranular twins eventually vanished after 50 h of aging. When the aging time was raised to 200 h, the incoherent twin structures nearly vanished, leaving just a small amount of coherent twin residue in the grains.

2.  In 316 LNSS treated with 750 °C aging, the second-phase $M_{23}C_6$ carbide will precipitate in the grains and grain boundaries, and with the increase in aging time, the $M_{23}C_6$ carbide will precipitate in the grain boundaries in the form of rods and chains, and eventually exhibit a network distribution.

3.  The rate of decline of room temperature Charpy impact energy in 316 LNSS drops continuously with increasing aging time, and the reduction is greater in the early aging period. However, the rate of decline of Brinell hardness tends to slow down as the aging time is extended.

4.  The impact energy and HBW are negatively linearly associated in the test data after aging, therefore the obtained HBW value may be utilized to predict the thermal embrittlement performance of 316 LNSS under real-life situations.

**Author Contributions:** Conceptualization, K.F.; methodology, K.L.; formal analysis, K.F.; investigation, K.F.; data curation, K.F.; writing—original draft preparation, K.F., K.L. and L.W.; writing—review and editing, K.F. and L.W.; supervision, K.F.; funding acquisition, K.F., K.L. and L.W. All authors have read and agreed to the published version of the manuscript.

**Funding:** This research received no external funding.

**Institutional Review Board Statement:** Not applicable.

**Informed Consent Statement:** Not applicable.

**Data Availability Statement:** Not applicable.

**Acknowledgments:** The authors would like to thank the National Key Research and Development Program of China (No. 2016YFB0700404) and Natural Science Foundation of China (No. 51375182) for their financial support.

**Conflicts of Interest:** The authors declare no conflict of interest.

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
