# Peer review of "Effect of Microstructure on Mechanical Properties of 316 LN Austenitic Stainless Steel"

_coatings, doi:10.3390/coatings12101461_

Round 1
Reviewer 1 Report
1. The existing title's uppercase and lowercase should be modified to conform to MDPI format.
2. The abstract section should include quantitative results.
3. Please give a “take-home” message as the conclusion of your abstract.
4. Keywords should be reordered based on alphabetical order.
5. Please use lowercase font for each of the keywords in accordance with MDPI format. Except for “316 LNSS;” and “HBW”.
6. It is unclear whether the author's something new in this work. According to evaluation, several published studies by other researchers in the past adequately explain related to microstructure and/or mechanical properties of Austenite stainless steel. Please be careful to highlight in the introduction section anything really innovative in this work.
7. The work, novelty, and limitations of similar prior studies must be explained in the introduction section to highlight the research gaps that the current study aims to fill.
8. Stainless steel have been widely used in the mechanical component of implant devices, especially total hip prosthesis. It is a crucial issue that authors should provide in the introduction and/or discussion section. The MDPI's suggested reverence should be adopted as follows: Computational Contact Pressure Prediction of CoCrMo, SS 316L and Ti6Al4V Femoral Head against UHMWPE Acetabular Cup under Gait Cycle. J. Funct. Biomater. 2022, 13, 64. https://doi.org/10.3390/jfb13020064
9. To help the reader grasp the study's workflow more easily, the authors could include more visuals to the materials and methods section in the form of figures rather than sticking with the text that now predominates.
10. Other information about the tool, such as the manufacturer, country, and specifications, should be provided.
11. The revised manuscript after peer review must provide detailed information on the error and tolerance of the experimental equipment utilized in this study. Due to the disparate outcomes of other researchers' subsequent studies, it would make for a valuable discussion.
12. An analysis of the findings with similar past research is required.
13. At the end of the discussion part, the present study's limitations must be added.
14. Add more detail to the conclusion by structuring it as a paragraph rather than in point-by-point as a present form.
15. The conclusion section needs to explain further research.
16. The reference should be enriched with literature from the last five years. Literature published by MDPI is strongly recommended.
17. The authors sometimes reduced a paragraph to just one or two phrases across the whole article, which made the explanation difficult to follow. To make a more thorough paragraph, the writers should expand upon their explanation. It is advised to include at least three sentences in a paragraph, one of which should serve as the primary idea and the others as supporting details. For example, line 211-216.
18. Because of grammatical faults and linguistic style, the authors must proofread the document. MDPI English editing service would be a solution.
19. Kindly double-check that the authors followed the MDPI format correctly, then modify the current form and recheck for any additional issues that have been discovered.
Author Response
On behalf of the authors, I would like to thank the reviewers for all the valuable comments which has helped us a great deal to improve the quality of our paper. We have answered all the queries raised by the reviewers to the best of our abilities and incorporated them into the revised manuscript. The modified parts have been highlighted in the revised manuscript. Once again, we would like to express our special gratitude for your careful review job. Please let us know if the revised paper now satisfies all requirements for publication. Please find our response to the comments raised by the Reviewers below.
1. The existing title's uppercase and lowercase should be modified to conform to MDPI format.
Thanks for your comment. The title's uppercase and lowercase have been modified to conform to MDPI format.
2. The abstract section should include quantitative results.
Thanks for your comment. The abstract has been revised as follows, The microstructure development of 316 LN austenitic stainless steel (316 LNSS) during the ag-ing process is investigated in this article. The thermal aging processes were conducted at 750 °C with different periods ranging from 50 to 500 hours. The metallographic results show that the coherent and incoherent twins were present in the original 316 LNSS grains, but dwindled away as the aging period increased. After 50 hours of aging, many fine, dispersed particles precipitat-ed from the matrix, which were identified as M23C6 by SEM-EDS and TEM. Additionally, the im-pact toughness and Brinell hardness (HBW) changed during the aging, which was closely related to the effects of dispersion strengthening and solution strengthening. A negatively linear rela-tionship between Brinell hardness and Charpy impact energy was established, which could be utilized to predict the degree of thermal embrittlement.
3. Please give a “take-home” message as the conclusion of your abstract.
Thanks for your comment. The abstract has been revised.
4. Keywords should be reordered based on alphabetical order.
Thanks for your comment. The order of keywords has been changed.
5. Please use lowercase font for each of the keywords in accordance with MDPI format. Except for “316 LNSS;” and “HBW”.
Thanks for your comment. The font has been changed.
6. It is unclear whether the author's something new in this work. According to evaluation, several published studies by other researchers in the past adequately explain related to microstructure and/or mechanical properties of Austenite stainless steel. Please be careful to highlight in the introduction section anything really innovative in this work.
Thanks for your comment. There is presently a dearth of research on the formation of precipitates in 316 LNSS at high temperatures since the bulk of previous studies have concentrated on the thermal deformation and dynamic recrystallization characteristics of this material at high temperatures. Furthermore, there is no discussion of the impact of short thermal aging at 750℃ on the organization's structure and characteristics; instead, the present focus of 316316 LNSS application performance study is on the organization's long-term thermal aging process at 300-400℃ base material and weld temperatures. This study examines the link between the impact toughness and Brinell hardness of 316LNSS as a result of thermal aging precipitates at 750℃ in an effort to forecast the status of impact toughness deterioration of real components.It is set out in lines 49 to 62 of the text.
7. The work, novelty, and limitations of similar prior studies must be explained in the introduction section to highlight the research gaps that the current study aims to fill.
Thanks for your comment. The tensile fracture behavior of 316 LNSS at 750 °C was not investigated by He et al.
[1], who used tensile experiments to investigate the thermoplastic and high temperature tensile fracture behavior of this stainless steel at 850-1300℃. The results showed that dynamic recrystallization occurred in this steel during thermal deformation above 1000℃. Jia et al.
[2] investigated the 316 LNSS thermal simulation testing apparatus. The results showed that at high temperatures, dynamic strain aging does place in 316 LNSS, but no connection between microstructure and dynamic strain aging behaviors was discovered.Samuel et al.
[3] examined how temperature and strain rate affected the tensile strength and section shrinkage of 316 LNSS at 750℃, but they did not examine how strain rate affected the material's mechanical properties. When 316 LNSS was subjected to thermal deformation, Mcqueen et al.
[4] studied the dynamic recrystallization behavior of the material and discovered that dynamic reversion and dynamic recrystallization could soften the metal and increase its plasticity.
However, they did not look into the deformation behavior or microstructure evolution of 316 LNSS. It is set out in lines 35 to 49 of the text.
[1].He, A.; Wang, X.T.; Xie G.L.; Yang X.Y.; Zhang, H.L. Modified Arrhenius-Type Constitutive Model and Artificial Neural Network-Based Model for Constitutive Relationship of 316 LN Stainless Steel During Hot Deformation. J. Iron. Steel Res. Int. 2015, 22(8), 721-729.
[2].Jia, S.G.; Tan, Q.H.; Ye, J.; Zhu, Z.W.; Jiang, Z.G. Experiments on Dynamic Mechanical Properties of Austenitic Stainless Steel S30408 and S31608. J. Constr. Steel Res. 2021, 179, 106556.
[3].Samuel, E.I.; Choudhary, B.K.; Bhanu Sankara Rao, K. Influence of temperature and strain rate on tensile work hardening behaviour of type 316 LN austenitic stainless steel. Scr. Mater. 2002, 46(7), 507-512.
[4].McQueen, H.J.; Yue, S.; Ryan, N.D.; Fry, E. Hot working characteristics of steels in austenitic state, J. Mater. Process. Technol. 1995, 53(1), 293-310.
8. Stainless steel have been widely used in the mechanical component of implant devices, especially total hip prosthesis. It is a crucial issue that authors should provide in the introduction and/or discussion section. The MDPI's suggested reverence should be adopted as follows: Computational Contact Pressure Prediction of CoCrMo, SS 316L and Ti6Al4V Femoral Head against UHMWPE Acetabular Cup under Gait Cycle. J. Funct. Biomater. 2022, 13, 64. https://doi.org/10.3390/jfb13020064
Thanks for this comment. The paper the reviewer mentioned has been cited in the revised manuscript.
9. To help the reader grasp the study's workflow more easily, the authors could include more visuals to the materials and methods section in the form of figures rather than sticking with the text that now predominates.
Thanks for this comment. In the section of materials and methods, the test process is relatively conventional and simple. Thus, the authors it is not necessary to all more visuals.
10. Other information about the tool, such as the manufacturer, country, and specifications, should be provided.
Thanks for this comment. The samples were examined with an optical microscope (OM, Zeiss Axiovert 200 MAT), and the precipitated phases were assessed with a spectrometer (Oxford Inca) and scanning electron microscope (SEM, Zeiss Merlin Compact), with the content of the precipitated phases being measured using the grid technique. And to examine the sample films, a transmission electron microscope (TEM, JEM-2100) was employed. It is set out in lines 81 to 93 of the manuscript.
11. The revised manuscript after peer review must provide detailed information on the error and tolerance of the experimental equipment utilized in this study. Due to the disparate outcomes of other researchers' subsequent studies, it would make for a valuable discussion.
Thanks for this comment. The test equipment used in the manuscript include metallographic microscopes, scanning electron microscopes, transmission electron microscopes, etc. These equipment are common internationally and the authors have marked their models in the manuscript, so that international colleagues can look up the error or tolerance of the equipment by the model number of the equipment.
12. An analysis of the findings with similar past research is required.
Thanks to catch them. The authors argue that the seventh and twelfth questions can be interpreted together.
13. At the end of the discussion part, the present study's limitations must be added.
Thank you for this suggestion. This has been modified in the manuscript from line 221 to 223.
14. Add more detail to the conclusion by structuring it as a paragraph rather than in point-by-point as a present form.
Thank you for this suggestion. The conclusions should be simple and clear. So the authors agree with the reviewer for this suggestion. And this has been modified in the manuscript.
15. The conclusion section needs to explain further research.
Thank you for this suggestion. This has been modified in the manuscript from lines 223 to 224.
16. The reference should be enriched with literature from the last five years. Literature published by MDPI is strongly recommended.
Thank you for this suggestion. This has been modified in the manuscript.
17. The authors sometimes reduced a paragraph to just one or two phrases across the whole article, which made the explanation difficult to follow. To make a more thorough paragraph, the writers should expand upon their explanation. It is advised to include at least three sentences in a paragraph, one of which should serve as the primary idea and the others as supporting details. For example, line 211-216.
Thanks to catch them. The author has changed some of the expressions in the manuscript.
18. Because of grammatical faults and linguistic style, the authors must proofread the document. MDPI English editing service would be a solution.
Thanks for your comment. We have corrected the grammatical faults and linguistic style.
19. Kindly double-check that the authors followed the MDPI format correctly, then modify the current form and recheck for any additional issues that have been discovered.
Thanks for your comment. We have corrected the form carefully.

Reviewer 2 Report
1- The main contribution of the work is not provided
2- The outcome of research must be present in the abstract such as percentage of improve or worst
3- Figure 3 , EDS result must present clearly and showed the point of the analysis
Author Response
On behalf of the authors, I would like to thank the reviewers for all the valuable comments which has helped us a great deal to improve the quality of our paper. We have answered all the queries raised by the reviewers to the best of our abilities and incorporated them into the revised manuscript. The modified parts have been highlighted in the revised manuscript. Once again, we would like to express our special gratitude for your careful review job. Please let us know if the revised paper now satisfies all requirements for publication. Please find our response to the comments raised by the Reviewers below. 1- The main contribution of the work is not provided Thanks for your comment. The specimens with different aging period suffered different thermal embrittlement corresponding to different hardness at room temperature. The thermal embrittlement is approximately evaluated with the hardness at room temperature. Although the power plant components is served at high temperature, the hardness measurements are usually conducted at room temperature. Therefore, the correlation between Charpy impact energy and hardness at room temperature is used in practical. 2- The outcome of research must be present in the abstract such as percentage of improve or worst The suggested reverence has been added,and the abstract has been revised. 3- Figure 3 , EDS result must present clearly and showed the point of the analysis Thanks to your suggestion, we have modified the presentation of Figure 3 and explained it in the article in lines 147-149.
Reviewer 3 Report
The manuscript deals with the option of using 316 LN austenitic steel (316LNSS) material for making components and pressure relief valves in ultra-supercritical (USC) units for enhanced power generation. The manuscript is well written with a logical explanation of the results. However, few observations were made which can be corrected or acknowledged by the authors for acceptance of the paper after the editor’s decision.
1) Kindly check the marking of the coherent and incoherent twins in Fig. 1b.
2) In line 177, it should be “time period” in place of “dates”.
3) Reframing the sentence starting from line 181 is recommended for better clarity of the reason.
4) The carbide percentage for each specimen exposed to different times was determined after the toughness test or before the test.
5) The phrase “Charpy impact energy” to Brinell hardness has been missed in line 211.
6) Determining the degree of thermal embrittlement using Charpy impact energy and hardness relation requires the determination of material hardness at different temperatures for forecasting. This idea should be present in the text.
Author Response
On behalf of the authors, I would like to thank the reviewers for all the valuable comments which has helped us a great deal to improve the quality of our paper. We have answered all the queries raised by the reviewers to the best of our abilities and incorporated them into the revised manuscript. The modified parts have been highlighted in the revised manuscript. Once again, we would like to express our special gratitude for your careful review job. Please let us know if the revised paper now satisfies all requirements for publication. Please find our response to the comments raised by the Reviewers below. 1) Kindly check the marking of the coherent and incoherent twins in Fig. 1b. Thanks to catch them. Fig. 1b has been checked and revised. 2) In line 177, it should be “time period” in place of “dates”. Thank you for this suggestion. The word ‘dates’ has been revised as ‘time period’. 3) Reframing the sentence starting from line 181 is recommended for better clarity of the reason. Thanks to catch them. The text has been revised as, ‘The plastic fracture feature with low grain boundary fraction could be attributed to the intergranular carbides. Under this condition, the carbide concentration was only approximately 3.19% as presented in Table 1.’ 4) The carbide percentage for each specimen exposed to different times was determined after the toughness test or before the test. Thanks to catch them. The carbide percentage for each specimen exposed to different times was determined after the toughness. 5) The phrase “Charpy impact energy” to Brinell hardness has been missed in line 211. Thanks to catch them. This has been revised. 6) Determining the degree of thermal embrittlement using Charpy impact energy and hardness relation requires the determination of material hardness at different temperatures for forecasting. This idea should be present in the text. Thank you for this suggestion. The specimens with different aging period suffered different thermal embrittlement corresponding to different hardness at room temperature. The thermal embrittlement is approximately evaluated with the hardness at room temperature. Although the power plant components is served at high temperature, the hardness measurements are usually conducted at room temperature. Therefore, the correlation between Charpy impact energy and hardness at room temperature is used in practical.
Round 2
Reviewer 1 Report
Excellent work.